# Evaluation of the Spray Drying Conditions of Blueberry Juice-Maltodextrin on the Yield, Content, and Retention of Quercetin 3-d-Galactoside

**DOI:** 10.3390/polym11020312

**Published:** 2019-02-13

**Authors:** María Z. Saavedra-Leos, César Leyva-Porras, Laura A. López-Martínez, Raúl González-García, Joel O. Martínez, Isaac Compeán Martínez, Alberto Toxqui-Terán

**Affiliations:** 1Coordinación Académica Región Altiplano, Universidad Autónoma de San Luis Potosí, Carretera Cedral Km, 5+600 Ejido San José de las Trojes Matehuala, S.L.P. C.P. 78700, México; zenaida.saavedra@uaslp.mx (M.Z.S.-L.); isaac.compean@uaslp.mx (I.C.M.); 2Centro de Investigación de Materiales Avanzados S.C. (CIMAV), Miguel de Cervantes No. 120, Complejo Industrial Chihuahua, Chihuahua, Chih. C.P. 31136, México; 3Coordinación Académica Región Altiplano Oeste, Universidad Autónoma de San Luis Carretera Salinas—Santo Domingo 200, Salinas de Hidalgo C.P. 78600, S.L.P. C.P. 78700, México; araceli.lopez@uaslp.mx; 4Facultad de Ciencias Químicas, Universidad Autónoma de San Luis Potosí, Av. Dr. Manuel Nava 6, San Luis Potosí, S.L.P. C.P. 78210, México; raulgg@uaslp.mx (R.G.-G.); atlanta126@gmail.com (J.O.M.); 5Centro de Investigación de Materiales Avanzados (CIMAV-Mty), Alianza Norte No 202, Parque de Investigación e Innovación Tecnológica, Apodaca Nuevo León C.P. 66600, México; alberto.toxqui@cimav.edu.mx

**Keywords:** Quercetin 3-d-galactoside, Analysis of variance (ANOVA), Spray Drying, Blueberry juice-maltodextrins, carrying agents

## Abstract

The influence of the processing conditions during the spray drying of mixtures of blueberry juice (BJ) and maltodextrin (MX) was determined quantitatively by the analysis of variance (ANOVA), and qualitatively by the surface response plots (SRP). The effect of two independent variables (inlet temperature, and MX concentration), and one categorical variable (type of MX), was determined on the yield (Y), content (Q), and retention (R) of the antioxidant quercetin 3-d-galactoside. From the ANOVA results, the concentration was the main variable affecting Y and Q, while temperature had a negligible effect, and the low molecular weight MXs exhibited a better response. The physicochemical characterization showed that the powder appearance and microstructure remained unaffected, but size and morphology of the particles varied with the processing conditions. This study established the optimal processing conditions for the spray drying of BJ-MX, and the application limits of the MXs based on the molecular weight distribution.

## 1. Introduction

The regular consumption of food with a high content of antioxidants is considered as an alternative to prevent chronic degenerative illnesses such as cancer, diabetes mellitus, cardio and cerebrovascular diseases, among others. The function of antioxidants is the neutralization of harmful body molecules in the form of free-unstable radicals [1,2,3]. The antioxidant activity is based on the presence of phenolic and flavonoid compounds such as quercetin 3-D-galactoside, resveratrol, myricetin, and kaempferol [4,5]. Some of the most important sources of antioxidants are berries, especially the raspberry, mulberry, strawberry, Chilean wineberry (*maqui*), and blueberry. Although the popularity in the consumption of blueberries has been extended to other regions of the world such as northern Europe, Asia, and Latin America, in 2014, the USA and Canada were the main producers and consumers of blueberries, with 90% of the total global production, equivalent to 444,814 tons [6,7]. The relatively short harvest season of blueberries, and the perishability of the fruit (with a water content of 85%), limit the availability in the market. Thus, more than 50% of the total production must be processed in the form of non-perishable products that may support a long shelf life such as juices, nectars, yogurts, marmalades, syrups, and juice powders [6,8]. Unfortunately, most of the processes include treatments at temperatures higher than 60 °C, which induce thermal degradation of the antioxidants [9,10]. Cano-Chauca et al. (2005) concluded that different blueberry juice processing methods might induce losses of 20%–50% of the antioxidant content [11]. For this reason, it is necessary to improve the methods for obtaining powders of blueberry juice, that preserve the nutritive and organoleptic characteristics, while reducing the loss of antioxidants. In this sense, the spray drying process is a viable alternative because it is economic and of easy operation [12,13].

The spray drying process is characterized by the use of relatively low temperatures and short residence times, which are suitable features for the drying of temperature-sensible products, allowing the preserving of some food properties such as flavor, color, odor, and nutrients [14]. The appearance of the obtained dried product is a fine powder with particles of regular shape and size, which is easy to handle, store and transport [11]. However, one of the difficulties that may arise during the spray drying of fruit juices is the caking of the product on the walls of the dryer, and the agglomeration of the powder, both causing low yields, operation problems, and erroneous prediction of the quality of the product [15]. The property that is related to this issue, is the glass transition temperature (*T*_g_) resulting from the high content of low molecular weight sugars such as fructose, glucose, and sucrose. An alternative for avoiding this problem is increasing the overall *T*_g_ of the food product with the aid of high molecular weight additives as carrying agents [16]. 

Maltodextrins (MX) are polysaccharides derived from acid or enzymatic hydrolysis of the starch. Based on their physicochemical properties, such as dextrose equivalent (DE), molecular weight distribution (MWD), and *T*_g_, these compounds have been successfully used as carrying agents in the spray drying of fruit juices [17,18]. Saavedra-Leos et al. (2015) demonstrated that, depending on the degree of polymerization (DP) and MWD, the maltodextrins could be employed in different technological applications [16]. For example, low DP maltodextrins may be used in processes where keeping the microstructural stability at temperatures about 90 °C is needed, while high DP maltodextrins may be employed as an additive in low temperature processes such as volume enhancers, and viscosity promoters. Additionally, other researchers have employed maltodextrin as a carrier agent in proportions of 10%–75% in the spray drying of different fruit juices [19,20,21]. Araujo-Díaz et al. (2017) studied the conservation of two antioxidants during the spray drying of blueberry juice, and compared the aid of two carrying agents (inulin and maltodextrin) [22]. They concluded that a maltodextrin with a DE of 10 showed a better performance than inulin, in the conservation of antioxidants, such as quercetin 3-d-galactoside. The spray drying conditions exerted were an inlet temperature of 180 °C, maltodextrin content of 30%, and feed flowrate of 7 mL/min. Recently, da Costa et al. (2018) reported the optimization of the spray drying conditions in the conservation of cupuassu (*Theobroma grandiflorum* Schum.) [23]. They found that the content of polyphenols, and flavonoids was maximized with an inlet temperature of 170 °C, maltodextrin concentration of 50%, and feed flowrate of 5 mL/min. However, no additional information related to the chemical structure of the maltodextrin was provided, i.e., DE, DP or MWD.

Thus, it is of vital importance to set the optimal drying conditions of the blueberry juice that may preserve the content of antioxidants. Therefore, a d-optimal experimental design, analysis of variance (ANOVA), and surface response methodology were employed in this work, in the optimization of the spray drying conditions of blueberry juice assisted with maltodextrin as a carrying agent. The effect of inlet temperature, maltodextrin concentration, and type of maltodextrin were evaluated in the powder yield, content and retention of quercetin 3-d-galactoside. The optical appearance, microstructure and particle size of the powder samples were characterized, and the results correlated with the processing variables. Additionally, this work demonstrates the application limits of maltodextrins when used as carrying agents in a technological process such as the spray drying of sugar-rich systems.

## 2. Materials and Methods

### 2.1. Materials

The juice was prepared using fresh blueberry fruit (*Vaccinium corymbosum*), commercially available in a market center (Costco Wholesale Corp., San Luis Potosí, México). The fruits were stored in a refrigerator for 12 h, and crushed in the juice extractor, Turmix E-17 (Guadalajara, Mexico). The juice and bagasse were stored in a glass container inside the fridge. The separation of the bagasse and the juice was done by vacuum filtration with a paper filter Whatman NO. 4, which is used in the clarification of juices and wines. Blueberry juice (BJ) was stored in the refrigerator at 4 °C, and in darkness to avoid degradation of the antioxidants.

Four types of maltodextrins (MX) were used as carrying agents, and were identified according to the dextrose equivalent (DE) as commercial grade maltodextrin (Mc), DE 10 (M10), DE 20 (M20), and DE 40 (M40). The Mc dry powder was purchased from INAMALT (Guadalajara, Mexico), while the M10, M20 and M40 were from INGREDION México (Guadalajara, Mexico).

### 2.2. Experimental Design

With the d-optimal experimental design, it is possible to determine both the individual effects of the evaluated variables and their interactions. Without the need for many experimental runs, and keeping the confidence interval [24], this methodology ensures the optimal selection of the spray drying conditions for obtaining a high yield of powder, while maximizing the content of antioxidants. Two independent continuous variables and one categorical independent variable were tested: inlet temperature (T), maltodextrin concentration (C), and dextrose equivalent (DE), respectively. The minimum and maximum levels of these variables were: inlet temperature of 170–210 °C, maltodextrin concentration of 10–30 wt %, and dextrose equivalent of Mc-M40. The response variables were the powder yield (Y), and the content of quercetin 3-d-galactoside retained (R). The d-optimal experimental design consisted of 25 experiments, necessary to achieve a quadratic model in the quantitative factors. From these 25 experiments, 5 runs were repeated, i.e. runs 8, 10, 11, 12 and 14 were repeated with runs 21, 22, 23, 24, and 25, respectively. All the experiments were executed randomly.

### 2.3. Spray Drying

The dehydration of blueberry juice was carried out in a Mini Spray Dryer B290 (Buchi, Switzerland). Mixtures of blueberry juice with maltodextrin (BJ-MX) were introduced into the spray dryer at room temperature. As a drying vehicle, hot air was injected at a volumetric flow rate of 35 m^3^/h, and constant pressure of 1.5 bar. The rest of the processing conditions were varied as inlet temperature 170–210 °C, concentration of MX 10–30 wt %, and type of maltodextrin Mc-M40. The percentage of the yield (*Y*) was calculated according to equation 1, with the masses of the collected dry powder (*W_P_*), and liquid (*W_L_*) fed into the dryer:(1)$$Y\left(\%\right)=\frac{W_{P}}{W_{L}}\times 100$$

### 2.4. Quercetin 3-d-Galactoside Content, and Retention

The antioxidant content of quercetin 3-d-galactoside was quantified by high performance liquid chromatography (HPLC) with a Waters system (Waters Corp. Milford, MA, USA), equipped with a binary pump, an auto-injector (model 717), and a dual wavelength absorbance detector (model 2487). The analyses were carried out at room temperature, and a pH of 3.0. A constant flow rate of 1 mL/min solution of 50% acetonitrile–phosphoric acid was employed as the mobile phase. Detection was set at 306 nm. The chromatographic separation was done with an Agilent C-18 column (75 × 4.6 mm DI 3.5 μm). All data were analyzed with the Empower Pro software (Version 4.0, Waters Corp., Milford, MA, USA). For more details about the antioxidant quantification by HPLC, please refer to Appendix A. 

Quercetin 3-d-galactoside was evaluated in the original juice, and in the dry powders. The content of the antioxidant was expressed as the micrograms of quercetin 3-d-galactoside per gram of blueberry juice powder (μg/g). The percent of retention (*R*) of quercetin 3-d-galactoside was determined according to Equation (2):(2)$$R\left(\%\right)=\frac{Q_{P}\times 1.7\times 100}{Q_{j}}$$
where *Q_P_* is the content of quercetin 3-d-galactoside in the dry powder (in ppm), and *Q_j_* is the content of quercetin 3-d-galactoside in the fresh juice (31.45 ppm). The value of 1.7 is a factor related to the humidity content, calculated as the ratio of the humidity of the injected juice sample (100%), and the humidity of the dried sample after the extraction process with methanol (58.46%).

### 2.5. Statistical Analysis

The effects of the three factors and their interactions were evaluated with the analysis of variance (ANOVA). A quadratic model with second-order interactions and the main effects were used to explain the relationship between the given continuous variables as indicated in Equation (3):(3)$$Z=\alpha_{0}+\sum \alpha_{i}X_{i}+\sum \alpha_{ii}X_{ii}^{2}+\sum \alpha_{ij}X_{i}X_{j}$$
where, *Z* represents the response variables (yield, or content of quercetin 3-d-galactoside), *X_i_*, *X_j_* are the factors (temperature and concentration of the maltodextrin), and *α_i_*, *α_ii_* and *α_ij_* are the linear regression coefficients of the model. 

In the process of selecting the model, the parameters of the complete model were first adjusted with Equation (2). Based on the normality test of Anderson Darling for the response variables (Y, and R), the transformation of the response was made by the Box–Cox analysis when it was necessary to stabilize the variance. Then, for simplification, the model was hierarchically pruned, and used only with the significant factors. Here we present the results obtained with the pruned model, and transformed into the response variables.

### 2.6. Physicochemical Characterization

The collected powders were optically characterized with a digital camera EOS Digital Rebel XSi, (Canon, Tokyo, Japan) with a 1:2.8 aspect EX/Sigma lens, and a high magnification digital camera THOUGH TG-5 operated in the microscope mode, (Olympus, Hamburg, Germany). The X-ray diffraction (XRD) analyses were carried out in an X’Pert Empyrean diffractometer (PANalytical, The Netherlands) with Cu-K radiation (λ = 1.5406 Å) operated at 40 kV, 30 mA and equipped with a X’Celerator detector in a Bragg–Brentano geometry. Scans were performed in the 2θ range of 4°–40°, with step size of 0.017° and 30 s per step. The morphology of the particles was observed in a scanning electron microscope (SEM) JSM-6010/LA (JEOL, Tokyo, Japan), operated at 15 kV, and in low-vacuum mode at 30 Pa. Powder sample was manually dispersed on a double-side carbon tape, and images were acquired at 100×–500×.

## 3. Results

### 3.1. Physicochemical Characterization of the Dry Powders

Figure 1 shows couples of optical micrographs of the BJ-MX samples after the spray drying process. As observed, the structure of some samples (Figure 1A) collapsed during the drying, while other samples remained as non-agglomerated powders (Figure 1B–K). A non-collapsed structure was observed as well-dispersed materials in bulk, i.e., non-agglomerated particles, and with a light purple appearance. The effect of the processing variables was observed as slight variations in the color of the powders (left image in Figure 1), and in the size of the agglomerated particles (right image in Figure 1). Although some of the BJ-MX mixtures were successfully dried, there were differences observed macroscopically and caused by the variations in the processing. On the contrary, the collapsed structure showed a notorious dark purple color, a bright surface, and an overall volume contraction. These observations indicated the collapse of the microstructure, suggesting the crystallization of the low-molecular weight sugar content in the BJ. Three undesired characteristics may be presented during the processing, handling, and storing of food products: stickiness, agglomeration, and caking [13]. Any of these characteristics indicate that during the drying of the sample, a rapid release of water causes the matrix of the food product to be unable to support its own weight, thus collapsing on itself. Additionally, the high content of low molecular sugars in the juice depress the overall *T*_g_ value, causing the collapse of the structure at lower temperatures. To avoid these problems, carrying agents such as MX are employed [15]. However, there are limits where the MX may act properly in the conservation of the microstructure of the BJ.

Figure 2 shows the XRD patterns of BJ-MX samples after the spray drying process. Overall, the diffractograms showed a broad peak at low diffraction angles (about 2θ=17°), and the absence of well-defined peaks, indicating the preservation of an amorphous microstructure [16]. Even the collapsed sample (run 9) showed a similar diffractogram. Unlike other carrying agents based on fructose units such as inulin, MXs are composed of glucose chains, which tend to remain in the amorphous state after the drying process. The advantage of this state is that it presents a high viscosity, which constrains the molecular mobility, while preventing chemical and biochemical reactions [22]. Additionally, Saavedra-Leos et al. (2018) reported that the MXs did not crystallize with the adsorption of water, but presented only a phase change from amorphous dried powders into liquid saturation, and further liquid condensation [25]. Thus, this indicated that under certain spray drying processing conditions, the microstructure of the BJ-MX collapsed not by the crystallization process, but by the phase change experimented during the rapid release of water molecules. Additional information regarding the thermal behavior of some selected samples can be found in Appendix A. 

Figure 3 shows the micrographs acquired by SEM of the powders that remained without microstructure collapse. In general, the morphology and size of the particles varied with the processing conditions. For some runs (Figure 3A,B), particles were observed as individual particles well separated from each other, i.e., without coalescing, with a spherical morphology, a rough surface, and with different sizes. Other powders (Figure 3C,E–G,K) presented a combination of individual spheres and coalesced particles. Likewise, among these samples, the particle size varied with the processing conditions, in the following run order 6>15>12>16>25. Powders corresponding to runs 21, 22 and 24 (Figure 3h–J) showed a larger content of coalesced material and pseudo-spherical particles; the particle size was similar among these runs. The collapsed sample (run 9, Figure 3d), showed micrometric agglomerated particles with irregular morphology. The spherical morphology is typically observed in the powders of maltodextrins obtained by spray drying [26,27]. Additionally, this morphology suggests the conservation of the microstructure, and thus the amorphous state [22]. However, the coalesced particles with irregular morphologies, which are commonly observed in collapsed systems, also remained in the amorphous state, which was corroborated by the XRD results and optical micrographs. According to Ahmed et al. (2018), the processing conditions such as inlet temperature and MWD have a negligible effect on the physicochemical properties of spray-dried inulin, but they affect the microstructure and morphology of particles [28]. Evidently, the variations in the particle size and morphology were caused by the differences in the MWD of the MXs, and by the different processing conditions exerted during the spray drying.

### 3.2. Yield of Dry Powders, Content and Retention of the Antioxidant

Table 1 shows the detailed description of the experimental runs, the powder yield, the content, and retention of the antioxidant, obtained after the spray drying of the BJ-MX at the different processing conditions. Although some qualitative relations among the independent and categorical variables may be inferred from Table 1, the quantitative analysis of the effect of the processing variables on the yield and the content of the antioxidant, will be discussed in Section 3.3. Figure 4 shows the calculated yield and retention of the 25 runs, from which 18 experiments resulted in obtaining a dried powder. The calculated yields were in the range of 0.12%–10.77%, with an average of 5.74%. On the other hand, the content of the quercetin 3-d-galactoside for these 18 powders was in the range of 0.0–2.36 μg/g of dried powder, and an antioxidant retention of 4.76%–13.7%. The average retention of the runs was 9.15%. However, there is not a direct relation between yield and the retention of the antioxidant, since the calculated values were very scattered. This means that there can be obtained low yields with high retention, and high yields with low retention. Of course, a high yield with a high retention is desired. In this sense, the highest yield obtained was 10.77% with a retention of 8.77% (run 14), while the lowest yield was 0.12% with a null retention (run 2). The highest retention obtained was 13.7% with a yield of 7.68% (run 20). According to Lim, Ma & Dolan (2011), the yield and retention of anthocyanin content in cull blueberry were 76%–79%, and 562 μg/g of blueberry solids, respectively, with a mixture of 30% of blueberry solids in maltodextrin [29]. They mentioned that the exposition to heat, oxygen and light, are the main sources affecting the degradation of antioxidants. Wach et al. (2007) obtained 9.5 mg of quercetin 3-d-galactoside per gram of onion (*Allium cepa* L.) when the antioxidant was extracted with 40% (*v*/*v*) of methanol in water [30]. Araujo-Diaz et al. (2017) determined a concentration of quercetin 3-d-galactoside of 0.093 ppm in a mixture of blueberry juice and 30% of maltodextrin [22]. The values of the yield reported herein may be relatively low when compared with those reported in the literature. However, it is worth mentioning that the yield was calculated based on the ratio of the masses of the dry powder, and the liquid mixture of BJ-MX fed into the dryer (Equation (2)).

### 3.3. ANOVA and Surface Response Methodology Analysis

The analysis of variance (ANOVA) was employed to determine the quality of the data, and to define the contribution of each of the independent variables on the response variables. The response surface methodology (RSM) was employed to observe the minimum and maximum variations of the data in the tested intervals. Table 2 shows the results of the ANOVA calculated for both response variables, yield and content of antioxidant. The F-value indicates the extent of the effect of the independent variable on the response variable, i.e. if F is equal to 1, then there is no effect, and as F increases above 1, the independent variable has a larger effect on the response variable. In this sense, the probability (p-value) of an F-value large enough to influence the experiment also indicates whether the independent variable affects the experiment. If the p-value is equal or less that the significance level, then the assumption of the influence of the variables on the experiments is correct. The significance level represents the probability of rejecting the previous assumption even if it was true. From Table 2, it is observed that temperature has a negligible effect on the yield, while the concentration of MX was the variable with the most important effect. The type of maltodextrin is important, but quantitatively its effect is less than that from the concentration. The p-value confirmed these observations for a significance level of 0.05. The interactions between the same variable (intra) showed that temperature (T^2^) has no effect on the yield, while the concentration (C^2^) has an effect, but less than the single concentration. The interaction between the variables (inter) showed that only the concentration and the type of maltodextrin (C·MX) had an effect on the yield, but the degree of the combined effects was still less than the single effect of concentration. Therefore, concentration was the independent variable with the larger effect on the yield.

The ANOVA results for the content of the antioxidant showed a similar behavior from those of the yield. The effect of the independent variables showed a major dependence on the concentration, rather than the temperature, and the type of maltodextrin. Concentration (C) was the only independent variable with an F-value relatively high, and a p-value below the significance value. The *inter*-type interaction of T·MX showed a positive F value, and p-value slightly above the significance value. For this reason, the effect of the T·MX interaction must be considered with restrictions, since its variance is to some extent away from the mean. The *intra*-type interactions of the variables showed that the effect of C^2^ was less than the single concentration, and its corresponding p-value was below the significance level.

For both response variables, yield and content of antioxidant, the inlet temperature (T) showed almost a negligible effect with a p-value above the confidence value in all the cases. This may indicate that the selected interval of temperatures was not suitable for the experiment, and another range of temperatures should have been selected, i.e., a wider range of inlet temperatures. However, this observation could not be known until the end of the experiment and the subsequent analysis of variance.

Figure 5 and Figure 6 show the corresponding response surface plots (RSP) for both, the yield and content of the antioxidant, as a function of the MX type. The effect of the processing variables on the yield as a function of the type of MX are presented in Figure 5A–D. As observed in the plots, the maximum in the yield was found for a concentration of 25% of MX. Lower values of the concentration of MX produced insignificant yield values. This behavior was similar for all the types of MXs, but the overall yield value decreased with the type of the MX, which was observed as a flat surface in the M20 and M40 RSP. Thus, the low molecular weight MXs (Mc, and M10) produced higher yields than the high molecular weight MXs (M20, and M40). Saavedra-Leos et al. (2015) characterized a set of maltodextrins similar to those employed herein, and reported an increase in the MWD and DP of the MXs as Mc<M10<M20<M40 [16]. Additionally, Wang et al. (2000), studied the branching of commercial grade MXs, and found that the higher the molecular weight, the higher the branching of the main chain of polymer [31]. Evidently, the branching is the physicochemical feature limiting the applicability of the MXs, where the carbohydrate polymer chains in the low molecular weight MXs are less branched, than in the high molecular weight MXs. A lower branching level indicates that polymer chains interact more readily with the functional groups from the BJ, i.e., less stearic hindrance. The effect of temperature on the yield showed a different behavior with the type of MX, since M10 and M40 showed an increment of about 5% when temperature increased from 170 to 210 °C, the Mc showed the opposite behavior, while in the M20 the yield remained almost constant with the temperature.

The RSP for the content of the quercetin 3-d-galactoside (Figure 6A–D) showed a more magnified behavior than that of yield. However, the Mc presented the highest content of the antioxidant at 25% of concentration of MX and 170 °C, while in the M10 the highest content value was about 5% at the same concentration value of 25% but at 210 °C. In both, M20 and M40, the content of the antioxidant was relatively low in the range of 2%–3%, and the effect of temperature was negligible.

Clearly, the results of the RSP helped to visualize the relation between the processing conditions, and the type of MX. Based on these observations, it was possible to elucidate the optimal conditions for obtaining the highest yield of powder with the highest content of the antioxidant. These conditions were a concentration of 25% of MX and 170 °C when employing the Mc, and 210 °C with the M10. Additionally, the RSP allowed setting the utilization limits of the MXs. Results showed that the low molecular weight MXs (Mc and M10) presented a better performance as carrying agents in the spray drying of BJ, while the higher molecular weight MXs (M20 and M40) were almost unresponsive in this application. Specifically, the type of MX showed a selective influence on the content of the antioxidant, since at low temperature the Mc had the highest value, while at higher temperature, the M10 showed a better performance. Clearly, these results demonstrated that the different MXs might be employed selectively as carrying agents in the spray drying of diverse processes. 

Table 3 summarizes the predictive equations extrapolated from the SRP for the yield and content of antioxidant as a function of the type of MX. The Box–Cox analysis revealed that a logarithmic transformation was necessary in order to stabilize the variance. It is worth mentioning that these equations are valid only within the range of conditions tested in this work. Overall, the drying temperature showed a negative effect on the yield while the concentration presented a positive effect. The positive values of the interactions between temperature and concentration (T·C), and the square of temperature (T^2^) seemed to be almost insignificant, but numerically their contributions were in the order of that from the concentration. The square of the concentration (C^2^) showed a negative effect, but less than the single concentration. The values of these three interactions (T·C, T^2^, and C^2^), showed constant values for all the types of MXs, suggesting the unresponsiveness of these interactions to the type of MX. Numerically, the effect of the type of MX showed that the Mc produced a higher yield at low temperatures, i.e., 170 °C, while in the M10 the highest yield was achieved at a higher temperature, i.e., 210 °C. The other two MXs (M20 and M40) were almost insensible to the changes in the variables, confirming the observed from the SRP. 

In general, the content of the antioxidant followed a similar behavior to that of the yield. The temperature had a negative effect on the content, while the influence of the concentration was positive and relatively larger than the former variable. The effect of the interactions (T·C, and T^2^) was positive and negative, respectively, and relatively lower than that of the concentration. The square of the concentration was negative, with a contribution in the order of that from the concentration. 

## 4. Conclusions

The effect of the processing variables on the spray drying of mixtures of blueberry juice (BJ) and maltodextrins (MX) were studied in this work. Two independent variables (inlet temperature and concentration of MX), and one categorical variable (type of MX) were tested in a d-optimal experimental design. The influence of the variables on the yield, content and retention of an antioxidant (quercetin 3-d-galactoside), was determined quantitatively by the analysis of variance (ANOVA), and qualitatively by the surface response plots (SRP). The results showed that the concentration was the main variable affecting both, the yield and content of the antioxidant, while temperature had a relatively low effect. Additionally, the low molecular weight MXs showed a better response to this technological application. The physicochemical characterization by optical micrographs, and X-ray diffraction (XRD), showed that the powder appearance, and microstructure, remained unaffected with the variations in the processing conditions. However, from the scanning electron microscopy (SEM) analysis, variations in both size and morphology of the particles were observed. Based on the SRP, a set of empirical equations were determined, which could be employed for predicting the yield and content of the antioxidant. With these results, it was possible to set the optimal processing conditions for the spray drying of BJ-MX, and elucidate the application limits of the MXs based on the molecular weight distribution. The commercial grade MX (Mc) showed optimal yield and content of the antioxidant quercetin 3-d-galactoside at a concentration of 25% of MX and a processing temperature of 170 °C, while the optimal performance of maltodextrin M10 was at 25% and 210 °C. The other two MXs tested (M20 and M40) were almost insensitive for this application. 

## Figures and Tables

**Figure 1 polymers-11-00312-f001:**
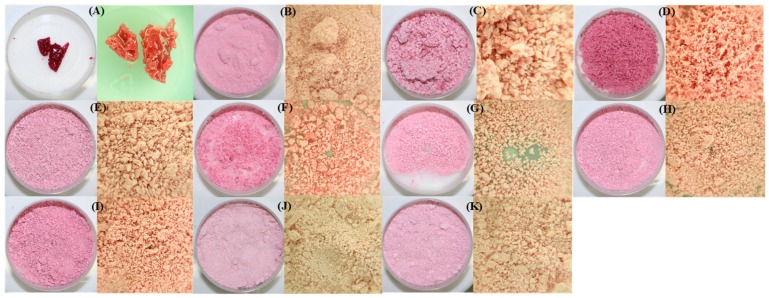
Optical micrographs of Blueberry juice-Maltodextrin (BJ-MX) samples after the spray drying process. Couples of images were arranged as low magnification in the left and high magnification in the right. The cylindrical container has a diameter of 4 cm, while the length of the high magnification image is 1 cm. Run identification (**A**) 9, (**B**) 3, (**C**) 5, (**D**) 6, (**E**), 12, (**F**) 15, (**G**) 16, (**H**) 21, (**I**) 22, (**J**) 24, and (**K**) 25.

**Figure 2 polymers-11-00312-f002:**
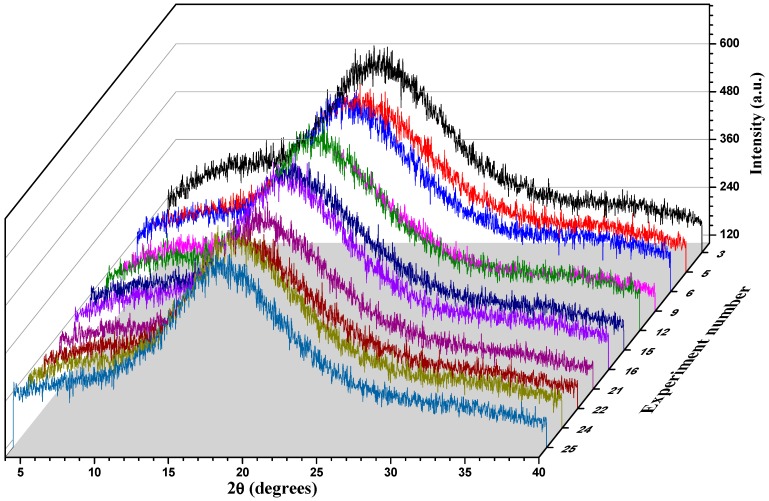
XRD patterns of BJ-MX samples after the spray drying process.

**Figure 3 polymers-11-00312-f003:**
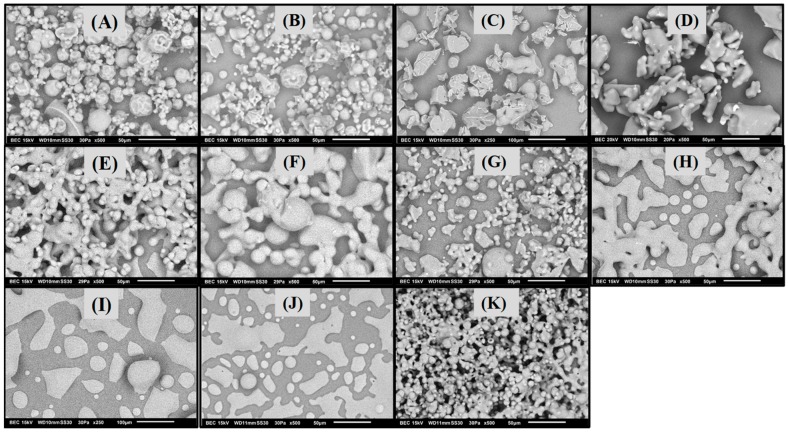
SEM micrographs of BJ-MX samples after the spray drying process. Run identification (**A**) 3, (**B**) 5, (**C**) 6, (**D**) 9, (**E**), 12, (**F**) 15, (**G**) 16, (**H**) 21, (**I**) 22, (**J**) 24, and (**K**) 25. Scale bar is equal to 50 μm in all the images, except in (**C**) and (**I**), where is equal to 100 μm.

**Figure 4 polymers-11-00312-f004:**
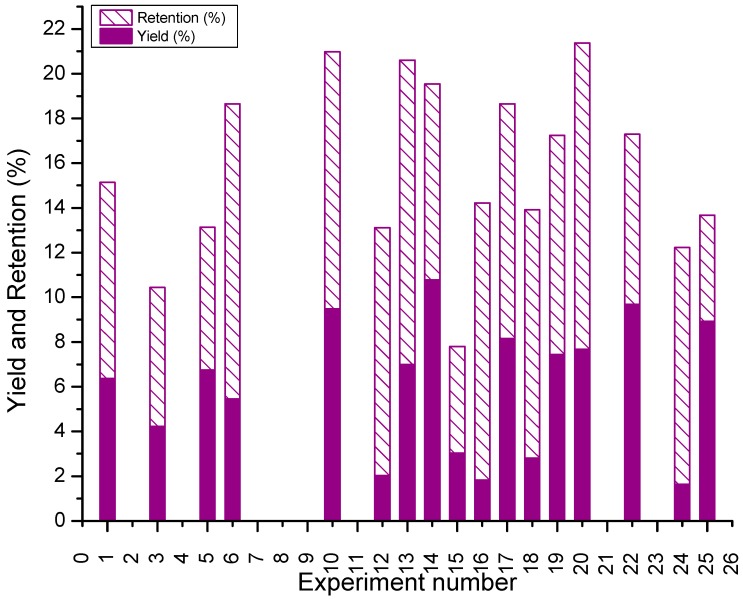
Yield (%), and retention (%) of quercetin 3-d-galactoside obtained for the 25 runs. The missing bars indicate the runs where sample collapsed during the spray drying process.

**Figure 5 polymers-11-00312-f005:**
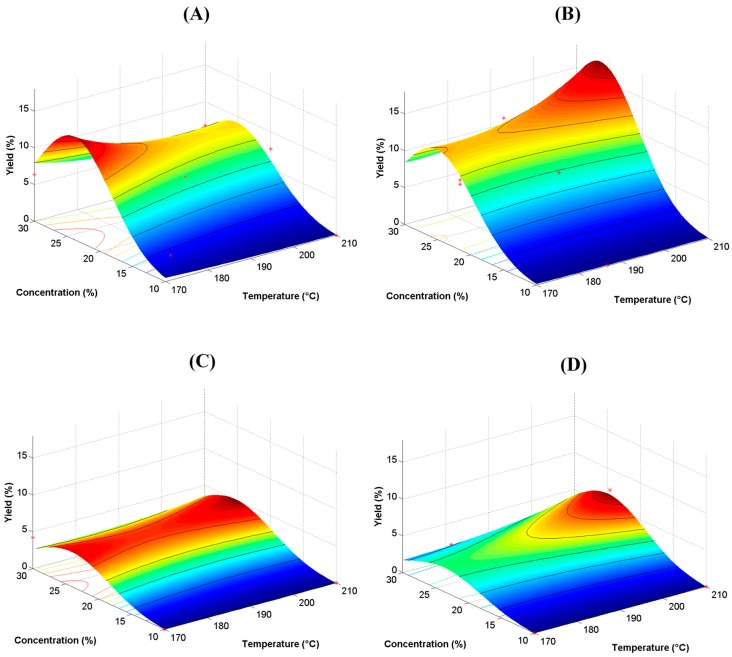
Response surface plots (RSP) for the yield of dry powder. Yield is expressed in percent of recovery (%): (**A**) Mc, (**B**) M10, (**C**) M20, and (**D**) M40.

**Figure 6 polymers-11-00312-f006:**
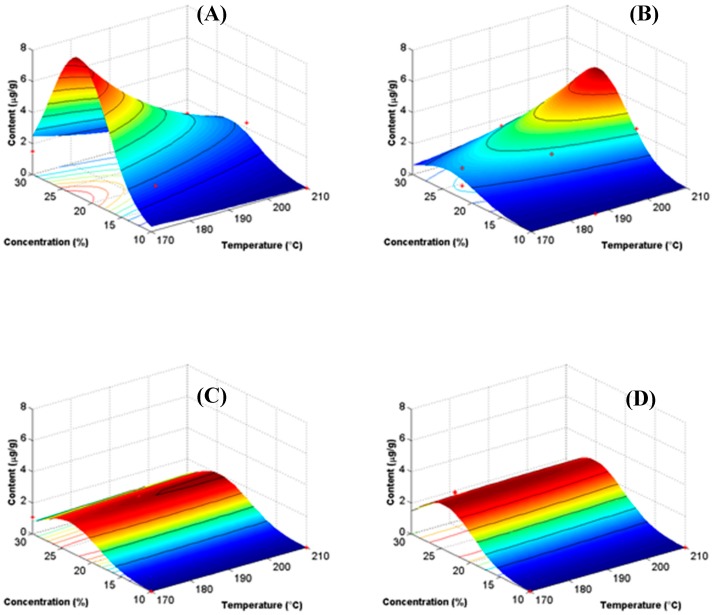
Response surface plots for the content of quercetin 3-d-galactoside in the dry powder. Content is expressed in micrograms of antioxidant per gram of dry powder (μg/g). (**A**) Mc, (**B**) M10, (**C**) M20, and (**D**) M40.

**Table 1 polymers-11-00312-t001:** Experimental results of dry powder yield, content and retention of quercetin 3-d-galactoside.

Experiment Number	Variation Levels of Variables			Dry Powder	Quercetin 3-d-Galactoside	
	T (°C)	C (%p/p)	MD	Yield (%)	Content (μg/g)	Retention (%)
1	170.0	30.0	Mc	6.37	1.51	8.77
3	170.0	30.0	M20	4.23	1.07	6.21
8	170.0	10.0	M40	0.00	0.00	0.00
10	170.0	21.6	M10	9.48	1.98	11.5
11	170.0	10.0	M20	0.00	0.00	0.00
21	170.0	10.0	M40	0.00	0.00	0.00
23	170.0	10.0	M20	0.00	0.00	0.00
25	170.0	21.6	M10	8.92	0.82	4.76
16	175.0	12.5	Mc	1.82	2.13	12.4
12	181.3	30.0	M40	2.02	1.92	11.1
24	181.3	30.0	M40	1.63	1.82	10.6
2	186.6	10.0	M10	0.12	0.00	0.00
13	190.0	20.0	Mc	7.00	2.35	13.6
17	190.0	19.6	M10	8.15	1.81	10.5
18	190.0	25.0	M20	2.81	1.91	11.1
14	193.1	30.0	M10	10.77	1.51	8.77
22	193.1	30.0	M10	9.68	1.31	7.61
19	209.4	24.2	M40	7.43	1.69	9.81
4	210.0	10.0	M40	0.00	0.00	0.00
5	210.0	30.0	Mc	6.74	1.1	6.39
6	210.0	18.4	M10	5.45	2.27	13.2
7	210.0	10.0	M20	0.00	0.00	0.00
9	210.0	10.0	Mc	0.00	0.00	0.00
15	210.0	30.0	M20	3.03	0.82	4.76
20	210.0	20.0	Mc	7.68	2.36	13.7

**Table 2 polymers-11-00312-t002:** ANOVA results determined for the yield and content of antioxidant in the BJ-MX.

	Yield					Content				
Source	DF	SS ^a^	MS ^b^	F	p ^c^	DF	SS ^a^	MS ^b^	F	p ^a^
**Model**	14	62.215	4.444	49.242	0.0000	14	85.414	6.101	20.914	0.0000
**T**	1	0.013	0.013	0.14	0.7158	1	0.350	0.350	1.198	0.2993
**C**	1	30.857	30.857	341.913	0.0000	1	41.530	41.530	142.362	0.0000
**MX**	3	3.717	1.239	13.729	0.0007	3	2.350	0.783	2.686	0.1032
**T·C**	1	0.295	0.295	3.268	0.1007	1	0.240	0.240	0.821	0.3861
**T·MX**	3	0.587	0.196	2.168	0.1550	3	2.966	0.989	3.389	0.0621
**C·MX**	3	0.350	0.117	1.293	0.3299	3	1.559	0.520	1.781	0.2142
**T^2^**	1	0.067	0.067	0.739	0.4100	1	0.001	0.001	0.004	0.9501
**C^2^**	1	7.877	7.877	87.283	0.0000	10	15.021	15.021	51.491	0.0000
**Residual**	10	0.902	0.090			5	2.917	0.292		
**Total**	24	63.118				24	88.33			

^a^ Sum of squares; ^b^ Mean squares; ^c^ Calculated at a significance level of 0.05; T = Temperature, C = Maltodextrin concentration, MX = Type of maltodextrin.

**Table 3 polymers-11-00312-t003:** Predicting equations extrapolated from the SRP for the yield and content of antioxidant in the BJ-MX, as a function of the type of MX.

TYPE OF MX	YIELD
**Mc**	$\ln\left(Y+0.2\right)=12.4315-0.1723T+0.5884C+0.0009T\cdot C+0.0004T^{2}-0.0154C^{2}$
**M10**	$\ln\left(Y+0.2\right)=8.3470-0.1542T+0.6239C+0.0009T\cdot C+0.0004T^{2}-0.0154C^{2}$
**M20**	$\ln\left(Y+0.2\right)=9.1970-0.1603T+0.5938C+0.0009T\cdot C+0.0004T^{2}-0.0154C^{2}$
**M40**	$\ln\left(Y+0.2\right)=6.5865-0.1450T+0.5801C+0.0009T\cdot C+0.0004T^{2}-0.0154C^{2}$
	**CONTENT**
**Mc**	$\ln\left(Q+0.03\right)=-0.8171-0.0358T+0.8108C+0.0008T\cdot C-0.00005T^{2}-0.0213C^{2}$
**M10**	$\ln\left(Q+0.03\right)=-15.6519+0.0311T+0.8816C+0.0008T\cdot C-0.00005T^{2}-0.0213C^{2}$
**M20**	$\ln\left(Q+0.03\right)=-10.1509+0.0013T+0.8761C+0.0008T\cdot C-0.00005T^{2}-0.0213C^{2}$
**M40**	$\ln\left(Q+0.03\right)=-9.9870-0.0015T+0.9058C+0.0008T\cdot C-0.00005T^{2}-0.0213C^{2}$

Y = Yield (%), Q = Content of antioxidant (μg/g), T = Temperature (°C), C = Concentration of maltodextrin (wt. %).

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
