# Peer review of "Evaluation of the Spray Drying Conditions of Blueberry Juice-Maltodextrin on the Yield, Content, and Retention of Quercetin 3-d-Galactoside"

_polymers, 2019, doi:10.3390/polym11020312_

Round 1
Reviewer 1 Report
The paper “Evaluation of the spray drying conditions of blueberry juice-maltodextrin on the yield, and retention of quercetin 3-D-galactoside” by Saavedra-Leos et al. investigates the optimal conditions for the spray drying conditions of blueberry juice in the presence of maltodextrin by statistical analysis.
The paper is well written, clear, and the conclusions are supported by the results. However, some modifications can be made in order to improve the overall quality of the manuscript.
1. The authors state that the agglomeration of the powders can be a result of different crystallization degrees of the samples. In order to confirm this hypothesis, some DSC analysis must be performed. Also, by TGA analysis will be determined the water content of each sample.
2. Table 1 does not exist!
3. Table 2, which is in fact table 1, must be reorganized as a function of temperature or concentration as these are the two independent variables. This reorganization will aid the lecturer to better understand the different correlations between the parameters.
4. In the conclusion section, I think that the authors should recall the optimal conditions of the process.
Minor corrections:
5. Page 1, line 41: “one of the most important…” and not “more”
6. Page 2, line 46: “…and the perishability of the fruit…”
7. The scale is not visible in Fig 3.
8. Page 6, line 207: “… acquired by SEM…”
9. Page 6, line 237: I think that the average of the calculated yields (Fig 4) is higher than 1,67%
10. As a general remark: the term “without collapse” is not very academic. I think that the authors may better use the term “non-agglomerated powders”
In view of the above, I recommend the publication of this manuscript in Polymers after minor corrections.
Author Response
The paper “Evaluation of the spray drying conditions of blueberry juice-maltodextrin on the yield, and retention of quercetin 3-D-galactoside” by Saavedra-Leos et al. investigates the optimal conditions for the spray drying conditions of blueberry juice in the presence of maltodextrin by statistical analysis.
The paper is well written, clear, and the conclusions are supported by the results. However, some modifications can be made in order to improve the overall quality of the manuscript.
Question 1: The authors state that the agglomeration of the powders can be a result of different crystallization degrees of the samples. In order to confirm this hypothesis, some DSC analysis must be performed. Also, by TGA analysis will be determined the water content of each sample.
Answer: In accordance with the Reviewer´s comment, DSC and TGA analysis were carried out on selected samples i.e. powders of experiments 3, 6, and 9. The results were included in the supplementary materials as:
Figure S1, shows the TGA, and DSC curves for some selected BJ-MX samples. From the TGA it was observed a humidity content of 2-5 % at 125 °C, and the onset of the thermal degradation about 160 °C. The TGA showed the presence of several degradation events in the range of 160-425 °C, indicated by the changes in the slope of the curve. At 425 °C and above, the BJ-MX was fully carbonized.
The DSC showed three endothermic events, the first related with the evaporation of water in a temperature range of 50-125 °C, the second identified as the melting of low molecular weight sugars at 130-175 °C, and the third corresponding to the melting of melting of high molecular weight sugars such as the maltodextrins at 175-250 °C. Each of the two melting events was followed by the thermal degradation of the low and high molecular weight sugars, content in the BJ-MX, respectively. Anyhow, the XRD results clearly showed the absence of diffraction peaks, indicating that powders remained in the amorphous state.
(A) |
(B) |
Figure S1. Thermal analysis of selected BJ-MX samples. A) TGA and B) DSC.
Question 2: Table 1 does not exist!
Answer: Authors appreciate the observation. Certainly, this was an error. The correction was made in the manuscript.
Question 3: Table 2, which is in fact table 1, must be reorganized as a function of temperature or concentration as these are the two independent variables. This reorganization will aid the lecturer to better understand the different correlations between the parameters.
Answer: In accordance with the Reviewer, Table 1 was reorganized as a function of temperature. Additionally, the amount of significant figures was modified according to the suggestion of one of the authors. Table 1 was modified as:
Table 1. Experimental results of dry powder yield, content and retention of quercetin 3-D-galactoside.
Experiment number | Variation levels of variables | Dry powder | Quercetin 3-D-galactoside | |||
T (°C) | C (%p/p) | MD | Yield (%) | Content (μg/g) | Retention (%) | |
1 | 170.0 | 30.0 | Mc | 6.37 | 1.51 | 8.77 |
3 | 170.0 | 30.0 | M20 | 4.23 | 1.07 | 6.21 |
8 | 170.0 | 10.0 | M40 | 0.00 | 0.00 | 0.00 |
10 | 170.0 | 21.6 | M10 | 9.48 | 1.98 | 11.5 |
11 | 170.0 | 10.0 | M20 | 0.00 | 0.00 | 0.00 |
21 | 170.0 | 10.0 | M40 | 0.00 | 0.00 | 0.00 |
23 | 170.0 | 10.0 | M20 | 0.00 | 0.00 | 0.00 |
25 | 170.0 | 21.6 | M10 | 8.92 | 0.82 | 4.76 |
16 | 175.0 | 12.5 | Mc | 1.82 | 2.13 | 12.4 |
12 | 181.3 | 30.0 | M40 | 2.02 | 1.92 | 11.1 |
24 | 181.3 | 30.0 | M40 | 1.63 | 1.82 | 10.6 |
2 | 186.6 | 10.0 | M10 | 0.12 | 0.00 | 0.00 |
13 | 190.0 | 20.0 | Mc | 7.00 | 2.35 | 13.6 |
17 | 190.0 | 19.6 | M10 | 8.15 | 1.81 | 10.5 |
18 | 190.0 | 25.0 | M20 | 2.81 | 1.91 | 11.1 |
14 | 193.1 | 30.0 | M10 | 10.77 | 1.51 | 8.77 |
22 | 193.1 | 30.0 | M10 | 9.68 | 1.31 | 7.61 |
19 | 209.4 | 24.2 | M40 | 7.43 | 1.69 | 9.81 |
4 | 210.0 | 10.0 | M40 | 0.00 | 0.00 | 0.00 |
5 | 210.0 | 30.0 | Mc | 6.74 | 1.1 | 6.39 |
6 | 210.0 | 18.4 | M10 | 5.45 | 2.27 | 13.2 |
7 | 210.0 | 10.0 | M20 | 0.00 | 0.00 | 0.00 |
9 | 210.0 | 10.0 | Mc | 0.00 | 0.00 | 0.00 |
15 | 210.0 | 30.0 | M20 | 3.03 | 0.82 | 4.76 |
20 | 210.0 | 20.0 | Mc | 7.68 | 2.36 | 13.7 |
Question 4: In the conclusion section, I think that the authors should recall the optimal conditions of the process.
Answer: The following was added to the conclusions:
The commercial grade MX (Mc) showed optimal yield and content of the antioxidant quercetin 3-D-galactoside at a concentration of 25% of MX and a processing temperature of 170 °C, while the optimal performance of maltodextrin M10 was at 25% and 210 °C. The other two MXs tested (M20 and M40) were almost insensitive for this application.
Question 5: Page 1, line 41: “one of the most important…” and not “more”
Answer: Text was modified as indicated by the Reviewer.
Question 6: Page 2, line 46: “…and the perishability of the fruit…”
Answer: The word “periashable” was replaced by “perishability”.
Question 7: The scale is not visible in Fig 3.
Answer: The following was added in the caption of Figure 3:
Scale bar is equal to 50 μm in all the images, except in (C) and (I), where is equal to 100 μm.
Question 8: Page 6, line 207: “… acquired by SEM…”
Answer: Text was modified as indicated by the Reviewer.
Question 9: Page 6, line 237: I think that the average of the calculated yields (Fig 4) is higher than 1,67%.
Answer: Authors appreciate this observation, which was not easy to detect. The calculated average yield and range were corrected as:
The calculated yields were in the range of 0.12-10.77%, with an average of 5.74%.
Question 10: As a general remark: the term “without collapse” is not very academic. I think that the authors may better use the term “non-agglomerated powders”.
Answer: In agreement with the Reviewer, the term “without collapse” was replaced by “non-agglomerated powders” and by “a non-collapsed structure”.

Reviewer 2 Report
Dear Authors,
having read your manuscript, I appreciate all your work put to obtain these high quality results.
I have some questions to your studies and some remarks to be addressed.
- how were the juice samples filtered prior to the study?
- where and when were the fruits purchased? who authenticated them? how long were the samples stored prior to the tests?
- what was the concentration of quercetine galactoside in the fresh juice?
- have the authors performed the quantitative study of quercetin galactoside in the juice prior to the tests to see how the content of quercetin galactoside depended on the temperature and/or storage time? please, comment on that
- the authors write, that they are using a 30% solution of formic acid in water. how aout the other line? what was the gradient used?
- please, add to the manuscript a chromatogram obtained for the tested juice, which was the base for the quercetine galactoside determination
- please, add more information on how was the content of quercetin galactoside determined in the spray dried samples - what was the quantity of the powder used, how was it extracted?
also, for the untreated juice - was the juice extracted to dtermine the content of flavonoid, or the quantity was measured directly in the juice? was the sample filtered prior to hplc injection?
- please, improve the details on the preparation of calibration curve equation of the reference compound: how many points were prepared, which concentration range was prepared?
- please, go through the manuscript again - there are some stylistic errors which need to be corrected prior to the publication, e.g. line 73,87, 117
= line 213: i do not see the number 7 in the figure 3
- please, write the Latin names in italics (line 83, 101)
- in the materials and methods section, please, add city and country of the producers of the listed equipment, e.g. apparatus. Add the mdel number to the Canon EOS used in the studies.
Author Response
Dear Authors,
Having read your manuscript, I appreciate all your work put to obtain these high quality results.
I have some questions to your studies and some remarks to be addressed.
Question 1: how were the juice samples filtered prior to the study?
Answer: In accordance with the Reviewer, the following was added in the manuscript in section 2.1:
The separation of the juice from the pulp was done by vacuum filtration with a paper filter Whatman N0. 4, which is used in the clarification of juices and wines.
Question 2: where and when were the fruits purchased? who authenticated them? how long were the samples stored prior to the tests?
Answer: The following was added to the manuscript in section 2.1:
The juice was prepared using fresh blueberry fruit (Vaccinium corymbosum), commercially available in a market center (Costco Wholesale Corp.). The fruits were stored in a refrigerator by 12 hours, and crushed in a juice extractor Turmix E-17 (Guadalajara, Mexico). The juice and pulp were stored in a glass container inside the fridge.
Question 3: what was the concentration of quercetine galactoside in the fresh juice?
Answer: The initial concentration of the antioxidant in the fresh juice was 31.45 ppm. The following was included in the manuscript:
where QP is the content of quercetin 3-D-galactoside in the dry powder (in ppm), and QJ is the content of quercetin 3-D-galactoside in the fresh juice (31.45 ppm).
Question 4: have the authors performed the quantitative study of quercetin galactoside in the juice prior to the tests to see how the content of quercetin galactoside depended on the temperature and/or storage time? please, comment on that.
Answer: We have not done the study suggested by the Reviewer. However, we believe that his idea opens an opportunity for contributing in the field of the degradation of antioxidants. Indeed, we are considering to perform the study in the following months.
Question 5: the authors write, that they are using a 30% solution of formic acid in water. how about the other line? what was the gradient used?
Answer: Authors apologize by the error in the writing of this part of the manuscript. The text was corrected, describing the solution employed for the dilution of the sample and as the mobile phase. The detailed description of the sample preparation and HPLC analysis was included in the supplementary materials as section S2.
The following was modified in the manuscript in section 2.4:
A constant flow rate of 1 ml/min solution of 50% acetonitrile-phosphoric acid was employed as the mobile phase.
Question 6: please, add to the manuscript a chromatogram obtained for the tested juice, which was the base for the quercetin galactoside determination.
Answer: The authors apologize for not being able to include the chromatogram as the Reviewer indicated. Unfortunately, we had some technical problems with the computer where the HPLC data is stored. The only data we have available are the values of the injections for the calibration curves, and the experiments of the BJ-MX samples in an excel file. The data is available upon request if the Reviewer consider this necessary for clarifying any doubt.
Question 7: please, add more information on how was the content of quercetin galactoside determined in the spray dried samples - what was the quantity of the powder used, how was it extracted?
Answer: In agreement with the Reviewer, the detail of the sample preparation was included in the supplementary materials in section S2. The following was added:
For the quantification of the antioxidant content in the dry samples, 0.5 g of powder were dissolved in 0.5 ml of phosphoric acid (10% v/v in water), and 3 ml of methanol as extracting solvent. Solution was stirred by 5 minutes, and left to rest by 24 hours in darkness in order to maximize the extraction of the antioxidants.
Question 8: also, for the untreated juice - was the juice extracted to determine the content of flavonoid, or the quantity was measured directly in the juice? was the sample filtered prior to hplc injection?
Answer: All the samples (fresh BJ juice, and dried powders) were treated in the same way. Samples were first dissolved, and then extracted with methanol. As in the previous question, the following was added in the supplementary materials:
For the quantification of the antioxidant content in the dry samples, 0.5 g of powder were dissolved in 0.5 ml of phosphoric acid (10% v/v in water), and 3 ml of methanol as extracting solvent. Solution was stirred by 5 minutes, and left to rest by 24 hours in darkness in order to maximize the extraction of the antioxidants. After the time was elapsed, the solution was filtered in an Acrodisc filter (0.45 μm). The filtered solution was diluted with 200 μL of methanol, and then a constant volume of 10 μL was injected in the HPLC instrument. Injections were carried out by triplicate.
Question 9: please, improve the details on the preparation of calibration curve equation of the reference compound: how many points were prepared, which concentration range was prepared?
Answer: In agreement with the Reviewer, the description of the preparation of the calibration curves was included in the supplementary materials as:
The calibration curves were constructed employing a quercetin 3-D-galactoside HPLC grade standard (>97.0%, Sigma-Aldrich). A stock solution of 1000 μg/ml was prepared, and from this solution, several aliquots (0.01, 1, 5, 10, and 20 μg/ml) were prepared in order to obtain the calibration curve. The calibration curves were prepared the same day of the injection in the HPLC.
Question 10: please, go through the manuscript again - there are some stylistic errors which need to be corrected prior to the publication, e.g. line 73,87, 117
Answer: Authors appreciate the observation of these style errors. All of them were corrected in the manuscript.
Line 73: For example, low DP maltodextrins may be used in processes…
Line 87: However, none additional information related with the chemical structure of the maltodextrin was provided i.e. DE, DP or MWD.
Line 117: this methodology ensures the optimal selection of the spray drying conditions for obtaining a high yield of powder, while maximizing the content of antioxidants.
Question 11: line 213: I do not see the number 7 in the figure 3
Answer: Again, authors appreciate for the observation of this error. The manuscript was corrected as:
Likewise, among these samples, the particle size varied with the processing conditions, in the following run order 6>15>12>16>25
Question 12: please, write the Latin names in italics (line 83, 101)
Answer: Latins names were modified as suggested by the Reviewer.
Line 83: cupuassu (theobroma grandiflorum Schum.)
Line 101: (Vaccinium corymbosum)
Question 13: in the materials and methods section, please, add city and country of the producers of the listed equipment, e.g. apparatus. Add the model number to the Canon EOS used in the studies.
Answer: The data requested by the Reviewer was included were it was available for the instruments and materials employed in the research. The model of the Canon EOS was added in section 2.6:
The collected powders were optically characterized with a digital camera (Cannon, EOS Digital Rebel XSi) with a 1:2.8 aspect EX/Sigma lens.

Round 2
Reviewer 2 Report
Dear Authors,
thank you for having revised your manuscript.
Please, do not forget to introduce a big letter for the gender name in the Latin nomenclature: Theobroma, instead of theobroma
do the authors have any other crude data from HPLC which would confirm the performed analyses, e.g. some reports, or lists of peak areas from the chromatograms with their retention times?
Author Response
Dear Reviewer,
The authors acknowledge the comments made in this second round of the reviewing process. Fortunately, on this occasion we were able to answer the question related with the chromatograms. Two days after submitting the first revision of the manuscript, the technical problem with the computer was fixed. In the next section, you will find the answers to your comments.
REVIEWER 2:
Dear Authors,
Thank you for having revised your manuscript.
Question 1: Please, do not forget to introduce a big letter for the gender name in the Latin nomenclature: Theobroma, instead of theobroma.
Asnwer: Text was modified as indicated by the Reviewer (line 83). Authors acknowledge the Reviewer for this observation.
Question 2: do the authors have any other crude data from HPLC which would confirm the performed analyses, e.g. some reports, or lists of peak areas from the chromatograms with their retention times?
Answer: The following was added as text, figures (S2, S3, and S4), and table S1 in the Supplementary materials and the titles where included in the corresponding section of the manuscript (i.e. Supplementary materials) (lines 411-414):
Figure S2 shows the chromatograms obtained for a calibration curve of quercetin 3-D-galactoside. The elution time of the antioxidant was 3.1 min, while the mobile phase eluted at 0.92 min. The intensity of the peak at 3.1 min linearly increased with the concentration of the antioxidant. After the integration of the area under the curve of the elution peak at 3.1 min, the data were plotted versus the concentration. Figure S3 shows the corresponding calibration curve and the repetitions. On the plots, the corresponding equation and the R-squared value were included, indicating the repeatability, and the closeness of the experimental data to the regression line, respectively. Table S1 summarizes the experimental data obtained for each calibration curve and the corresponding repetitions. The average values from the linear regression (m, b and R2) were employed for calculating the concentration of the antioxidant in the dry powders. Figure S4 shows a typical chromatogram of an injected sample from a dry powder (experiment 18). Additionally to the mobile phase and antioxidant peaks, were observed other peaks at elution times of 1.5 and 2.5 minutes, which may correspond to the methanol added during the extraction process, and some other impurities contained in the juice. After integrating the area under the curve, the content of the antioxidant was determined from the comparison with the calibration curve.
Figure S2. Chromatograms obtained for the calibration curve of quercetin 3-D-galactoside at five concentrations (0.1, 1, 5, 10, and 20 μg/ml).
Figure S3. Calibration curves for quercetin 3-D-galactoside.
Table S1. Experimental data obtained for each calibration curve.
CONCENTRATION (μg/ml) |
AREA (a.u.) |
AVERAGE | STANDARD DEVIATION | ||
R1 | R2 | R3 | |||
0.1 | 1371.0 | 1276.0 | 1363.0 | 1336.67 | 52.69 |
1 | 13971.0 | 13837.0 | 14450.0 | 14086.00 | 322.27 |
5 | 72064.0 | 74283.0 | 75426.0 | 73924.33 | 1709.46 |
10 | 155466.0 | 156665.0 | 157336.0 | 156489.00 | 947.34 |
20 | 305756.0 | 316680.0 | 315546.0 | 312660.67 | 6006.44 |
Linear regression | |||||
m | 15386.6 | 15912.4 | 15838.5 | 15712.50 | 284.65 |
b | -1365.7 | -2339.3 | -1530.1 | -1745.03 | 521.17 |
R2 | 0.9998 | 0.9999 | 0.9999 | 0.9999 | 0.0001 |
Figure S4. Typical chromatogram of an injected dry powder (experiment 18) after the extraction process.
